# Rapid adaptation to malaria facilitated by admixture in the human population of Cabo Verde

Iman Hamid[1]*, Katharine L Korunes[1], Sandra Beleza[2], Amy Goldberg[1]*

[1]Department of Evolutionary Anthropology, Duke University, Durham, United States; [2]Department of Genetics and Genome Biology, University of Leicester, Leicester, United Kingdom

**Abstract** Humans have undergone large migrations over the past hundreds to thousands of years, exposing ourselves to new environments and selective pressures. Yet, evidence of ongoing or recent selection in humans is difficult to detect. Many of these migrations also resulted in gene flow between previously separated populations. These recently admixed populations provide unique opportunities to study rapid evolution in humans. Developing methods based on distributions of local ancestry, we demonstrate that this sort of genetic exchange has facilitated detectable adaptation to a malaria parasite in the admixed population of Cabo Verde within the last ~20 generations. We estimate that the selection coefficient is approximately 0.08, one of the highest inferred in humans. Notably, we show that this strong selection at a single locus has likely affected patterns of ancestry genome-wide, potentially biasing demographic inference. Our study provides evidence of adaptation in a human population on historical timescales.

**\*For correspondence:**
imanhamid95@gmail.com (IH);
amy.goldberg@duke.edu (AG)

**Competing interests:** The authors declare that no competing interests exist.

## Introduction

Genetic studies have demonstrated the important role of adaptation throughout human evolution, including classic examples such as loci underlying pigmentation, and adaptation to high-altitude life-styles and infectious disease (*Sabeti et al., 2002*; *Ohashi et al., 2004*; *Lamason et al., 2005*; *Voight et al., 2006*; *Norton et al., 2007*; *Nielsen et al., 2007*; *Pickrell et al., 2009*; *Yi et al., 2010*; *Fumagalli et al., 2011*; *Grossman et al., 2013*; *Lachance and Tishkoff, 2013*). Yet, we have a limited understanding of adaptation in human populations on historical timescales, that is, during the last tens of generations. The ongoing selective pressures shaping human genomic variation, and how quickly humans can adapt to strong selective pressures, remain unclear. Adaptation on these short timescales is of particular importance because large-scale migrations within the past thousands to hundreds of years have exposed human populations to new environments and diseases, acting as new selective pressures (*Hellenthal et al., 2014*; *Mathias et al., 2016*; *Busby et al., 2016*; *Patin et al., 2017*; *Laso-Jadart et al., 2017*; *Nielsen et al., 2017*; *Fernandes et al., 2019*).

Admixture—gene flow between previously diverged populations to form a new population with ancestry from both source populations—provides a particularly rapid opportunity for selection to act in a population by introducing alleles previously adapted in a source population into the admixed population (*Huerta-Sánchez et al., 2014*; *Jeong et al., 2014*; *Racimo et al., 2015*; *Norris et al., 2020*). Additionally, in recent human admixture, ancestry contributions from each source population are often large enough to introduce alleles at intermediate frequencies, potentially avoiding loss from drift (*Ruiz-Linares et al., 2014*; *Mathias et al., 2016*; *Fernandes et al., 2019*; *Fortes-Lima et al., 2019*). More generally, admixture is ubiquitous in human history (*Moorjani et al., 2011*; *Hellenthal et al., 2014*; *Busby et al., 2015*; *Triska et al., 2015*; *Busby et al., 2016*; *Laso-*

*Jadart et al., 2017*; *Patin et al., 2017*); therefore, understanding the effects of selection in these often understudied populations is essential to the study of human evolution.

The admixture process may obscure the signals commonly used to detect selection by increasing linkage disequilibrium and changing the distribution of allele frequencies (*Lohmueller et al., 2010*; *Lohmueller et al., 2011*; *Gravel, 2012*; *Racimo et al., 2015*). Further, common signatures of selection, such as deviation from neutral expectations of the allele frequency spectrum, may not be sensitive to adaptation on the scale of tens of generations (*Sabeti et al., 2006*; *Field et al., 2016*). Recent progress has given insights into human adaptation during the past few thousand years by using allele frequency trajectories from ancient DNA (*Lindo et al., 2016*) or the distribution of singletons in extremely large data sets (*Field et al., 2016*). We focus on admixed populations as an opportunity to detect rapid adaptation using modern populations and moderate sample sizes, allowing broader sets of populations to be studied. That is, within-genome ancestry patterns across multiple nearby loci may be easier to detect than single allele frequency shifts (*Tang et al., 2007*). Further, ancestry-based methods constrain the timing of potential selection to post-admixture, providing concrete information about the timing of selection, whereas non-ancestry-based summary statistics may detect selection in the source populations.

We test this concept developing new ancestry-based methods to characterize adaptation to malaria during the ~20 generations since the founding of the admixed human population of Cabo Verde. The Republic of Cabo Verde is an archipelago off the coast of Senegal and was uninhabited before settlement in ~1460 by Portuguese colonizers and enslaved peoples from the Senegambian region of West Africa (*Fernandes et al., 2003*; *Beleza et al., 2012*; *Verdu et al., 2017*; *Korunes et al., 2020*), henceforth referred to as 'European' and 'West African' source populations, respectively. This is approximately 19–22 generations ago assuming a 25- to 28-year human generation time (*Fenner, 2005*). Recent analyses using genetic ancestry information alongside historical data confirmed that admixture in Cabo Verde likely began within the last ~20 generations (*Korunes et al., 2020*). In this study, we assume admixture occurred 20 generations ago, and we focus on three major island regions of Cabo Verde: Santiago, Fogo, and the Northwest Cluster (*Figure 1A*).

The malaria parasites *Plasmodium vivax*, *P. falciparum*, and *P. malariae* have been reported across the islands of Cabo Verde since settlement; recurrent malaria epidemics have primarily occurred in highly populated regions (*World Health Organization et al., 2012*; *DePina et al., 2019*). Santiago, which has consistently been the most densely populated of the Cabo Verde islands, has experienced the most substantial burden of malaria transmission since settlement ~20 generations ago. Personal and historical accounts of malaria incidence within Cabo Verde described the largest and most populous island, Santiago, as the most 'sickly' and 'malarious' (*Patterson, 1988*). In the last century, malaria epidemics of both *P. vivax* and *P. falciparum* have occurred primarily in Santiago (*Snow et al., 2012*; *World Health Organization et al., 2012*; *Ferreira, 2017*; *DePina et al., 2018*). It is not fully understood why Santiago has sustained a higher burden of malaria than the other Cabo Verdean islands (*World Health Organization et al., 2012*); however, it may be due to a combination of higher population density, climatic differences between islands, increased migration into Santiago, which has historically served as the main trading port for Cabo Verde, and the suitability of the island for the mosquito vector. The other two island regions we consider share ancestry components with Santiago, but largely lacked the selective pressure of recurrent malaria transmission, providing a unique opportunity to compare related populations with and without malaria as a selective pressure.

We hypothesized that admixture has facilitated rapid adaptation to the malaria parasite *Plasmodium vivax* via the malaria-protective *Duffy antigen receptor for chemokines* (*DARC*) locus (also known as *Atypical Chemokine Receptor 1* [*ACKR1*]) in Santiago. The protective allele is almost fixed in West African populations and rare elsewhere (*Howes et al., 2011*; *Gething et al., 2012*). The malaria parasite *P. vivax* uses the chemokine receptor encoded by the *DARC* gene to enter and infect red blood cells. The Duffy-null allele (also known as FY*O, rs2814778) is protective against *P. vivax* infection via a single nucleotide polymorphism (SNP) that disrupts binding of an erythroid-specific transcription factor in the promoter region (*Mercereau-Puijalon and Ménard, 2010*; *Gething et al., 2012*). Thus, individuals carrying the null allele have reduced expression of Duffy antigens on the surface of the blood cell, protecting against *P. vivax* infection. Duffy-null is a classic example of strong selection in the human lineage, and it has been estimated to be under one of the

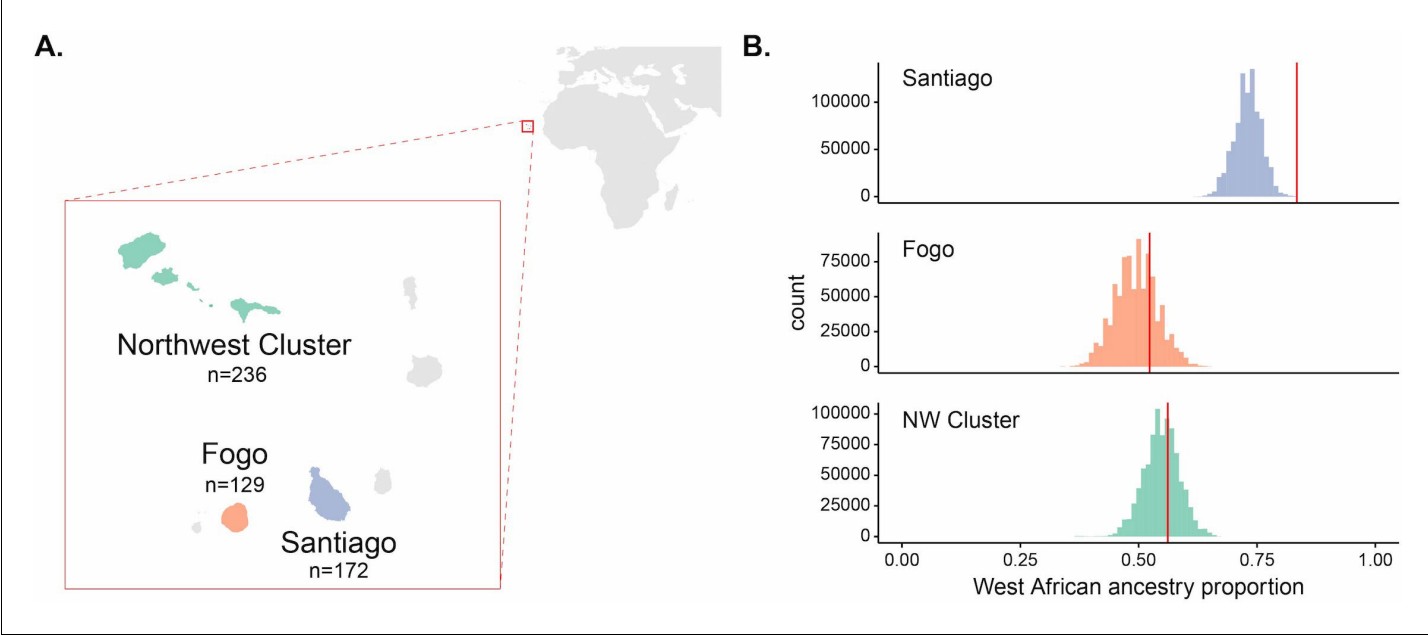

**Figure 1.** Enrichment of West African ancestry at the *DARC* locus in Santiago, Cabo Verde. (A) Map of Cabo Verde islands and sample sizes for number of individuals from each island region. (B) The distribution of West African-related local ancestry proportion across the genome by SNP (n = 881,279) by island, with the *DARC* locus marked by vertical red lines. Local ancestry was estimated using RFMix (see Materials and methods). The *DARC* locus is an outlier for high West African-related ancestry in Santiago, but not Fogo or the Northwest Cluster.

The online version of this article includes the following figure supplement(s) for figure 1:

**Figure supplement 1.** Local ancestry proportion along the genome in Santiago.

**Figure supplement 2.** The observed frequency of Duffy-null for each island vs neutral expectation based on mean global ancestry (as estimated by ADMIXTURE).

strongest selective pressures in human history (*Hamblin and Di Rienzo, 2000*; *Hamblin et al., 2002*; *Kwiatkowski, 2005*; *McManus et al., 2017*), suggesting it is a plausible selective pressure in Cabo Verde.

Interestingly, this hypothesis goes back to a voyage in 1721 in which Captain *Roberts, 1745* reported that a disease in Santiago is 'dangerous to strangers' during the rainy season. Consistent with ancestry-mediated protection from malaria, the record has been interpreted by medical historians to suggest that 'foreign visitors and residents of European descent seem to have suffered more than the African and Afro-Portuguese majority' from malaria in Santiago (*Patterson, 1988*).

In this study, we combine ancestry-based summary statistics and simulations to identify and characterize selection at the malaria-protective *DARC* locus on the island of Santiago during the ~20 generations since the onset of admixture. Importantly, we also consider the genome-wide consequences of this mode of selection. That is, we find that strong selection at a single locus may shift genome-wide ancestry patterns, with potential to bias demographic inference. The results of this study provide evidence for rapid adaptation in human populations and advance our ability to detect and characterize selection in recently admixed populations.

## Results

### Enrichment of West African ancestry at the *DARC* locus in Santiago

Empirical studies of selection in admixed populations often look for regions of the genome that deviate from genome-wide patterns of genetic ancestry (*Tang et al., 2007*; *Jin et al., 2012*; *Jeong et al., 2014*; *Rishishwar et al., 2015*; *Triska et al., 2015*; *Zhou et al., 2016*; *Busby et al., 2017*; *Laso-Jadart et al., 2017*; *Patin et al., 2017*; *Fernandes et al., 2019*; *Lopez et al., 2019*; *Norris et al., 2020*; *Vicuña et al., 2020*). Regions of the genome with substantially higher ancestry from one source than present on average in the rest of the genome are hypothesized to be enriched

for genes under selection. For an allele at different frequencies in the source populations, selection will increase the frequency of the ancestry on which the adaptive allele occurs at that locus. We tested if the *DARC* locus was an outlier within the genome for West African ancestry. We estimated local ancestry using the RFMix software (*Maples et al., 2013*) and calculated the proportion of individuals with West African ancestry at each SNP (see Materials and methods for details on local ancestry assignment). *Figure 1B* and *Figure 1—figure supplement 1* show the distribution of West African ancestry for over ~880,000 SNPs by island, with the value for the Duffy-null SNP position marked in red. Within Santiago, this locus has the highest frequency of West African ancestry in the population, occurring at 0.834 frequency compared to mean West African ancestry across SNPs for individuals from Santiago of 0.730. In contrast, the *DARC* locus is not an outlier in its frequency of West African ancestry on the other island regions, occurring at the 75th and 65th percentiles for Fogo and the NW Cluster, respectively. High West African local ancestry proportion at the *DARC* locus in Santiago is consistent with the expectation that the Duffy-null allele rapidly increased in frequency following admixture, simultaneously increasing the proportion of individuals with West African ancestry at that locus relative to the genome-wide average.

Under a simple population genetic model, we expect the frequency of a neutral allele in an admixed population to be a linear combination of the allele frequencies in each source population and their relative ancestry contributions. The Duffy-null allele is nearly fixed in the West African source population and largely absent in the European source population; therefore, under neutrality, the expected frequency of the allele in each Cabo Verdean population is approximately equal to the West African ancestry contribution. Using the observed global ancestry proportion inferred with ADMIXTURE (*Alexander and Lange, 2011*) as an estimate of the ancestry contribution from West Africa to each island, we found that the Duffy-null allele is at a higher frequency than expected under neutrality for the island of Santiago, but not the other regions of Cabo Verde (*Figure 1—figure supplement 2*, *Table 1*, binomial test, Santiago: $p = 2.193 \times 10^{-5}$; Fogo: $p = 0.1915$; NW Cluster: $p = 0.8172$).

## Long, high-frequency West African ancestry tracts span the *DARC* locus on Santiago

The distribution of the lengths of ancestry tracts spanning a selected locus can provide information for detecting and characterizing selection beyond single-locus outlier tests. In the case of recent admixture and strong selection, we might generally expect to see a parallel increase in local ancestry proportion in the regions surrounding the beneficial locus because the beneficial allele increases before recombination can break up large surrounding ancestry blocks. This is analogous to the increase in linkage disequilibrium and homozygosity in non-admixed populations (*Sabeti et al., 2002*; *Kim and Nielsen, 2004*; *Voight et al., 2006*). *Figure 2A* plots ancestry tracts that span the *DARC* locus for individuals from Santiago. As expected from source population allele frequencies, the Duffy-null allele is contained on all West African ancestry tracts spanning the locus and not found on any European ancestry tracts. Consistent with recent selection, West African ancestry tracts are longer and in higher frequency than European ancestry tracts covering the region. The median West African ancestry tract length spanning the locus is ~85 Mb, while the median European tract length is ~39 Mb.

In order to test if the observed local ancestry patterns are suggestive of selection beyond genome-wide ancestry proportion, we developed a summary statistic based on the length and

**Table 1.** Expected and observed Duffy-null allele frequencies for each island and source population.
Expected Duffy-null frequencies are approximated by mean West African global ancestry proportion for each island, calculated using the ADMIXTURE software.

| Population | n (sampled individuals) | Expected frequency | Observed frequency | Binomial test p-value |
|---|---|---|---|---|
| Santiago | 172 | 0.737 | 0.834 | $2.193 \times 10^{-5}$ |
| Fogo | 129 | 0.498 | 0.539 | 0.192 |
| NW Cluster | 236 | 0.552 | 0.557 | 0.817 |
| GWD | 107 | 0.997 | 1.000 | - |
| IBS | 107 | 0.002 | 0.019 | - |

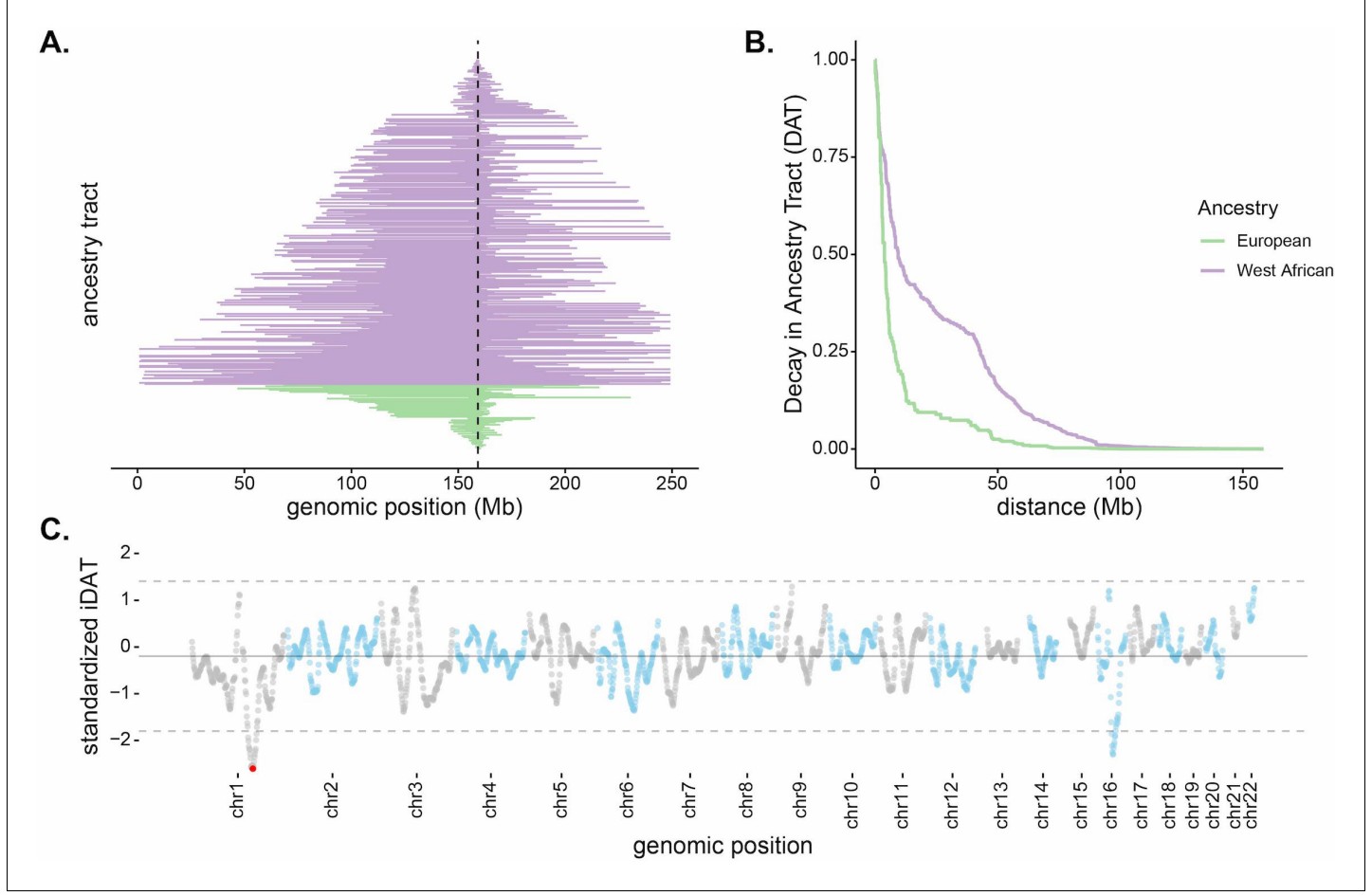

**Figure 2.** Long, high-frequency West African ancestry tracts span the *DARC* locus in Santiago. (**A**) The distribution of West African (purple) and European (green) ancestry tract lengths spanning the *DARC* locus (dashed line). Each horizontal line represents a single chromosome in the population (n = 343, one chromosome was excluded due to having unknown ancestry at the *DARC* locus). (**B**) Decay in Ancestry Tract (*DAT*) as function of absolute distance from the Duffy-null allele for West African (purple) and European (green) ancestry tracts. (**C**) Mean standardized integrated *DAT* (*iDAT*) score for 20 Mb sliding windows (step size = 1 Mb), using standardized *iDAT* for 10,000 random positions across the genome. Horizontal solid gray line indicates mean windowed standardized *iDAT* score (−0.196), and horizontal dashed gray lines indicate three standard deviations from the mean windowed score. The red dot is the most extreme windowed standardized *iDAT* score (−2.602), indicative of a larger area under the curve for West African *DAT* compared to European *DAT*. This 20 Mb window contains the Duffy-null SNP.

The online version of this article includes the following figure supplement(s) for figure 2:

**Figure supplement 1.** Mean standardized integrated Decay in Ancestry Tract (*iDAT*) score for 20 Mb sliding windows (step size = 1 Mb), using standardized *iDAT* for 10,000 random positions across the genome for (**A**) Fogo and (**B**) the Northwest Cluster.

**Figure supplement 2.** Density distributions for five ancestry-based statistics under eight neutral models.

**Figure supplement 3.** Density distributions for five ancestry-based statistics under simulations using different genetic maps.

**Figure supplement 4.** Performance of integrated Decay in Ancestry Tract (*iDAT*) under various scenarios.

**Figure supplement 5.** Performance of integrated Decay in Ancestry Tract (*iDAT*) for various chromosome sizes and cut-off values.

frequency of the tract-length surrounding a locus. The integrated Decay in Ancestry Tract (*iDAT*) score compares the rate of decay of ancestry tract lengths as a function of distance from a site of interest (*Figure 2B*; see Materials and methods for details and performance of *iDAT* statistic under various demographic scenarios). The statistic is analogous to the commonly used integrated haplotype score (*iHS*) (*Voight et al., 2006*), but considers local ancestry tracts instead of haplotypes. Negative values for *iDAT* indicate longer West African ancestry tracts at higher frequencies compared to European ancestry tracts. Positive values indicate longer European ancestry tracts at higher frequencies compared to West African ancestry tracts. Windows that contain multiple extreme values of *iDAT* provide stronger evidence for recent selection. *Figure 2C* plots *iDAT* values along the genome

for individuals from Santiago. Values are calculated by averaging over 20 Mb sliding windows (step size of 1 Mb) for 10,000 random standardized *iDAT* scores. Notably, in Santiago, the *DARC* gene is contained in the window with the lowest *iDAT* score in the genome (*Figure 2C*; window *iDAT* score = −2.602). *iDAT* scores near the *DARC* locus are not outliers in other island regions (*Figure 2— figure supplement 1*).

## Ancestry-based signatures for Santiago cannot be explained by drift alone

In order to estimate the expected distribution of ancestry within the population and test if the values of various summary statistics for the *DARC* locus on Santiago can be explained by drift alone, we conducted neutral simulations in SLiM (*Haller and Messer, 2019*) (Materials and methods). We calculated the following five summary statistics for each simulated population: the West African local ancestry proportion at *DARC*, the variance in the frequency of West African local ancestry across SNPs on chromosome 1, the median and mean West African ancestry tract length containing the Duffy-null allele, and the unstandardized *iDAT* score for the Duffy-null SNP. The variance in local ancestry along the chromosome provides a non-LD-based measure to capture high frequency and long tracts of West African ancestry using the population-wide measures of local ancestry for each SNP on the chromosome. Studies of demographic history and selection in recently admixed populations often assume constant population size and a single admixture event to simplify simulations. In order to confirm that assumptions about demographic history do not change our expectations, we considered multiple scenarios of population growth, differences in population size, and models of both constant contributions and single admixture events (*Table 2*). The values for the summary statistics for Santiago generally lie outside our expectations for all models, especially considered jointly (*Figure 2—figure supplement 2*).

Together, these summary statistics provide suggestive evidence that the *DARC* locus has been under positive selection on the island of Santiago since admixture started ~20 generations ago. To formally test this hypothesis, we extended the SWIF(r) framework developed by *Sugden et al., 2018*. SWIF(r) is a machine learning classification framework that explicitly learns the joint distributions for a set of features and returns a posterior probability of positive selection at a site of interest. It is particularly useful for handling summary statistics that are correlated, such as the length and frequency of ancestry tracts. We trained SWIF(r) using data simulated in SLiM and estimated the posterior probability of positive selection at the *DARC* locus using the five ancestry-based measures (Materials and methods). SWIF(r) returned a high posterior probability of positive selection at *DARC* on Santiago starting 20 generations ago ($P>0.999$).

## Classical haplotype-based signatures of selection not detected at the *DARC* locus

The haplotype-based statistic, *iHS*, is often used to detect signatures of recent positive selection and partial selective sweeps (*Voight et al., 2006*), particularly in non-admixed populations. This statistic has been used as evidence of selection in recently admixed populations (*Fernandes et al., 2019*; *Reynolds et al., 2019*; *Norris et al., 2020*). However, the process of admixture results in the

**Table 2.** Demographic models used for single-chromosome neutral simulations relevant to Cabo Verde demographic history.

| Initial population size (N) | Population growth model | Population growth rate (per generation) | Admixture type | Proportion of new migrants (per generation) | Scenario number |
|---|---|---|---|---|---|
| 1000 | Constant size | - | Single-pulse | - | 1 |
| | | | Continuous | 0.01 | 2 |
| | Exponential | 0.05 | Single-pulse | - | 3 |
| | | | Continuous | 0.01 | 4 |
| 10,000 | Constant size | - | Single-pulse | - | 5 |
| | | | Continuous | 0.01 | 6 |
| | Exponential | 0.05 | Single-pulse | - | 7 |
| | | | Continuous | 0.01 | 8 |

mixture of differentiated allele frequencies and diverged haplotypes, so interpretation of these statistics is difficult and applicability is limited. We demonstrate this by calculating *iHS* for all SNPs in our data set for each island region and performing the common standardization based on allele frequencies, using the software *hapbin* (*Maclean et al., 2015*). *Figure 3* shows the distribution of absolute standardized *iHS* values along the genome for each island population, with the Duffy-null SNP indicated by the orange dot and flag. The absolute *iHS* values for all islands at the Duffy-null SNP are low. That is, the commonly used statistic *iHS* does not detect significant signatures of selection at the Duffy-null SNP position. This analysis, and other summaries of variation that do not account for the allele frequency and LD changes associated with admixture, may be detecting the high diversity in the African source populations rather than post-admixture selection. Without considering the process of admixture, we should be skeptical of the utility of these statistics in recently admixed populations. This emphasizes the importance of new methods that are admixture-aware.

## Strong selection inferred at the *DARC* locus in Santiago

Beyond identifying selection, inference of the strength of selection is informative about the evolutionary processes shaping human genomes. We used two complementary approaches to infer the strength of selection at the *DARC* locus. First, we considered a deterministic classical population-genetic model of selection based on the trajectory of allele frequencies over time on a grid of possible dominance and selection coefficients (Materials and methods). The estimate of the selection coefficient depends on dominance; past studies have modeled Duffy-null as recessive (*Hodgson et al., 2014*), dominant (*Pierron et al., 2018*), and additive (*McManus et al., 2017*) when estimating selection strength in other human populations. *Figure 4A* plots the selection strength ($s$) as a function of the dominance coefficient ($h$) of the Duffy-null allele for a set of three realistic initial frequencies, assuming 20 generations of constant selection strength. Functional studies suggest that heterozygotes have at least partial protection against *P. vivax* infection (*Cavasini et al., 2007*; *Sousa et al., 2007*; *Gething et al., 2012*; *Kano et al., 2018*); while not an exact correlate for population-genetic model parameters, this suggests that the Duffy-null allele is unlikely to be fully recessive or fully dominant. Taking the mean of selection coefficients for $0.2 \leq h \leq 0.8$, we estimate the selection coefficient for each initial frequency, $s_{0.65} = 0.106$, $s_{0.70} = 0.082$, and $s_{0.75} = 0.056$, where $s_{p_o}$ is the inferred selection coefficient for initial allele frequency $p_o$.

Second, we used a simulation and rejection framework, approximate Bayesian computation (ABC), to jointly infer the selection coefficient and initial West African contribution while allowing for drift (Materials and methods). We used the five ancestry-based summary statistics described previously. We assumed an additive model, a single admixture event, and exponential growth in the population. Taking the median of the posterior distribution as the point estimate for selection coefficient, we estimated $s = 0.0795$ (*Figure 4B*; see *Figure 4—figure supplement 1* for estimates of $s$ when modeling Duffy-null as either a dominant or a recessive mutation). This estimate of selection coefficient is consistent with those estimated under the deterministic population-genetic model.

## Selection at a single locus impacts genome-wide ancestry estimates

Mean global ancestry proportion is often used as an estimate for initial ancestry contributions for admixed populations (*Moreno-Estrada et al., 2013*; *Hellenthal et al., 2014*; *Bryc et al., 2015*; *Mathias et al., 2016*; *Laso-Jadart et al., 2017*; *Patin et al., 2017*; *Fernandes et al., 2019*). However, our ABC estimates of the initial contributions from West Africa are lower than the mean ancestry currently observed in Santiago. The median of the posterior of initial contributions from West Africa is 0.690, with the middle 50 percentile of observed values in [0.682,0.697] (*Figure 5A*). In contrast, the observed mean ancestry in Santiago (*Figure 1B*) is 0.737. While this particular difference in observed and inferred founding contributions may be due to sampling biases or other neutral processes, it raises the more general question of how strong selection at a single locus may impact genome-wide ancestry patterns. We hypothesized that selection at *DARC* may have increased the genome-wide West African ancestry proportions in the current population of Santiago.

To test the genomic consequences of post-admixture selection at a single locus, we simulated whole human autosomes under a model of exponential growth and a single admixture event with selection at a single locus. We first considered a model based on the history of Santiago, using the posterior distributions of selection coefficient and initial West African ancestry contribution as

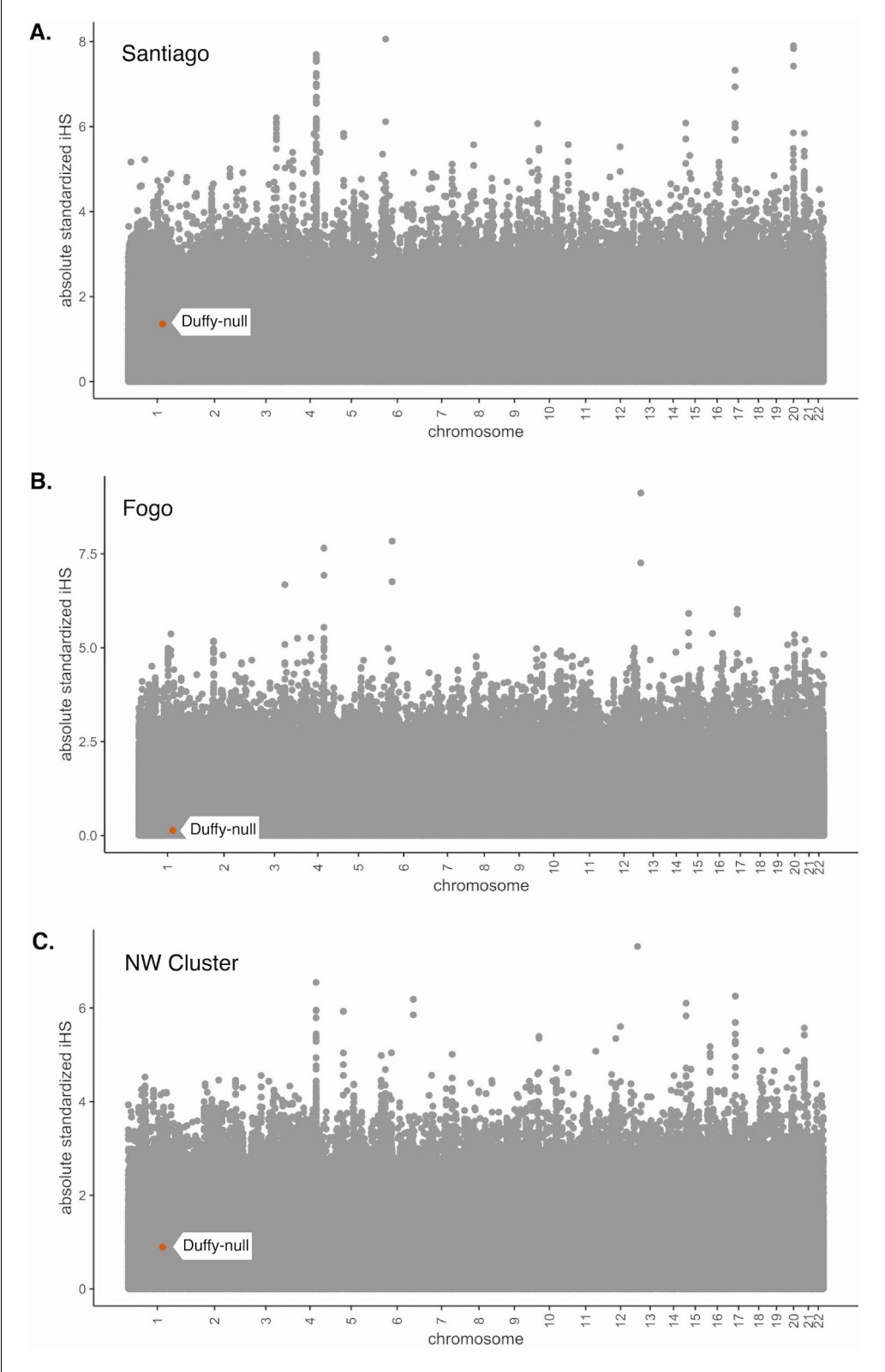

**Figure 3.** Absolute values of *iHS* for SNPs in the Cabo Verde data set. *iHS* was calculated using the *hapbin* software and standardized using the default method based on allele frequencies. (**A**) Santiago, (**B**) Fogo, and (**C**) NW Cluster. Value for Duffy-null SNP is indicated by orange dot and white label. Duffy-null *iHS* value is nonsignificant in all island regions.

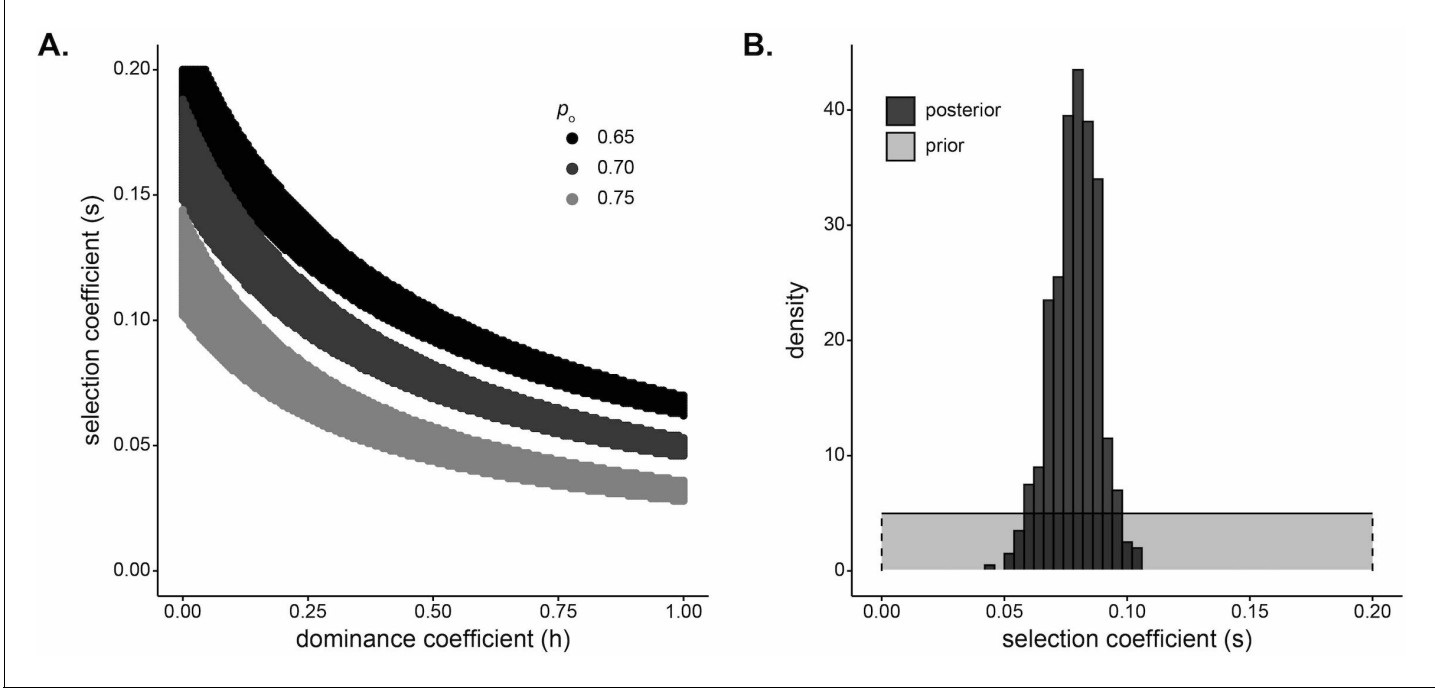

**Figure 4.** Strong selection inferred at the *DARC* locus in Santiago. (**A**) Pairs of *s* and *h* that result in a small difference in final allele frequency calculated under the model and the allele frequency observed in the Santiago genetic data, $|p_{20} - p_{Duffy}| < 0.01$ under a deterministic population genetic model. Colors indicate the initial Duffy-null frequency: $p_o = 0.65$, black; $p_o = 0.70$, dark gray; $p_o = 0.75$, light gray. (**B**) Approximate Bayesian computation (ABC) estimates of the selection coefficient for Duffy-null on Santiago. Shaded gray area shows prior distribution of selection coefficient [$s \sim U(0, 0.2)$]. Dark gray histogram shows posterior distribution for selection coefficient (median = 0.0795), constructed from regression-adjusted values from accepted simulations.

The online version of this article includes the following figure supplement(s) for figure 4:

**Figure supplement 1.** Results of approximate Bayesian computation (ABC) estimation of posterior distributions for (**A**) selection coefficient for Duffy-null and (**B**) initial West African ancestry contribution for Santiago.

**Figure supplement 2.** Results of leave-one-out cross-validation of approximate Bayesian computation (ABC) joint estimation.

parameters for the simulations (Materials and methods). *Figure 5A* plots the estimated posterior distribution of initial West African contribution in dark gray and the simulated distribution of global ancestry after 20 generations in white. The distribution of global ancestry in the populations simulated with selection (median 0.723, white) is noticeably higher than the initial contributions specified in the simulations (median 0.690, dark gray). This demonstrates that selection at a single locus is a plausible mechanism to increase mean global ancestry in an admixed population under a scenario similar to Santiago.

To explore the mechanism and relationship between selection strength at a single locus and genome-wide ancestry patterns, we simulated whole autosomes, assuming a single admixture event with initial West African ancestry contribution at 0.65 and selection coefficient varying from 0 to 0.2 at a single locus on chromosome 1. *Figure 5B* plots the mean ancestry for chromosome 1 and the other 21 autosomes in each simulated population after 20 generations as a function of the selection coefficient at a single locus. Perhaps surprisingly, mean ancestry on chromosomes 2–22 also increases with selection strength (gray), indicating that global ancestry increases beyond the contribution of higher ancestry on the selected chromosome alone (black). Together, this evidence suggests that strong selection at the *DARC* locus over 20 generations may have skewed global ancestry in Santiago and raises potential biases with a statistic that is often used to infer neutral demographic histories.

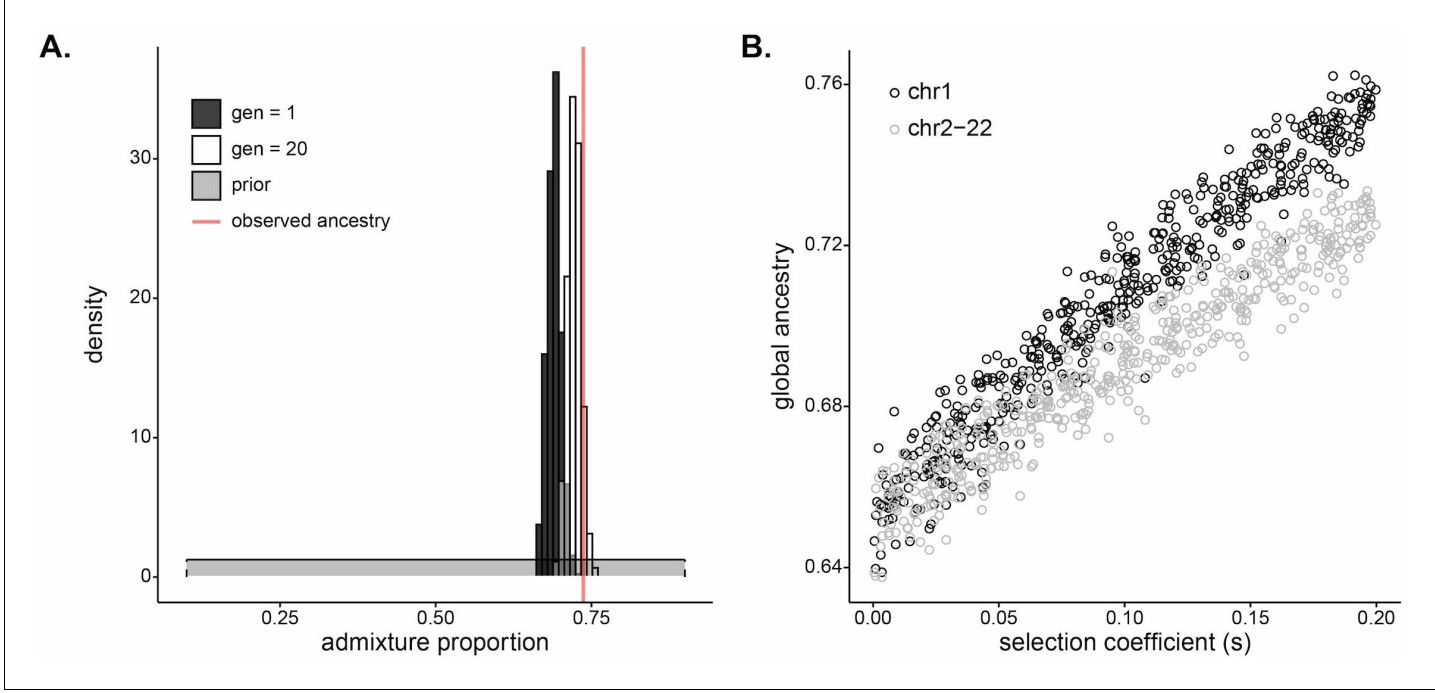

**Figure 5.** Selection at a single locus impacts genome-wide ancestry proportion. (A) Inferred (dark gray), simulated (white), and observed (red) mean of global ancestry in Santiago over time. The dark gray histogram plots the posterior distribution for initial ($g = 1$) West African ancestry contribution inferred using approximate Bayesian computation (ABC) (median, 0.690); the prior distribution [$m \sim U(0.1, 0.9)$] is in light gray. The red line plots the mean global ancestry estimated by ADMIXTURE from modern genetic data from Santiago, 0.737. The observed global ancestry is higher than most values of the initial contributions inferred in dark gray. The white histogram plots the distribution of West African global ancestry proportion calculated after 20 generations in populations simulated with selection coefficients and initial ancestries drawn from the ABC-inferred values (median, 0.723). The global ancestry calculated after 20 generations of simulated selection (white) more closely matches that observed from Santiago genetic data (red line). (B) West African mean global ancestry proportion calculated for 500 simulated populations after 20 generations under varying single-locus selection coefficients, $s$. We simulated whole autosomes, setting the initial West African ancestry contribution to 0.65. Black circles indicate mean ancestry on chromosome 1 alone. Gray circles indicate mean ancestry on the other autosomes (2–22). The increase in ancestry with selection for gray circles demonstrates that selection impacts global ancestry beyond the local effects of the chromosome under selection.

The online version of this article includes the following figure supplement(s) for figure 5:

**Figure supplement 1.** Effect of selection on global ancestry across simulation methods.

## Discussion

Using adaptation to malaria in the admixed population of Cabo Verde as a case study, we have demonstrated that admixture can facilitate adaptation in merely tens of generations in humans. Developing methods to identify and characterize post-admixture selection, we found that this rapid adaptation leaves detectable genomic signatures (*Figures 1* and *2*), with potential genome-wide consequences (*Figure 5*). Combining inference under two complementary methods, and under a range of possible dominance coefficients and initial allele frequencies, we estimated selection strength of $s \approx 0.08$ for the Duffy-null allele in Santiago (*Figure 4*). Our estimate is consistent with other studies that have inferred the strength of selection for Duffy-null ranging from ~0.04 (modeled under additive selection) in sub-Saharan African populations (*McManus et al., 2017*) and ~0.07 (modeled as recessive) to ~0.2 (modeled as dominant) in a Malagasy population with admixed African ancestry (*Hodgson et al., 2014*; *Pierron et al., 2018*). Our estimated strength of selection for Duffy-null is among the highest inferred for a locus in any human population.

Introgression of an adaptive allele can facilitate adaptation on short timescales, particularly for traits with large effects from single loci. When ancestry contributions from multiple sources are high, such as is common in recent human admixture, selection post-admixture can be a faster mode of adaptation, similar to selection on standing variation (*Hermisson and Pennings, 2005*; *Hedrick, 2013*).

Commonly used ancestry outlier approaches have identified candidate regions for admixture-enabled adaptation in many populations; for example, at the *HLA* and *LCT* regions in Bantu speaking populations (*Patin et al., 2017*), the *MHC* locus in Mexicans (*Zhou et al., 2016*), and adaptation to high-altitude at *EGLN1* and *EPAS1* in Tibetans (*Jeong et al., 2014*). However, results from this framework alone can be difficult to interpret because drift post-admixture may substantially change allele frequencies and the distribution of local ancestry within and between individuals (*Bhatia et al., 2014*; *Belbin et al., 2018*; *Calfee et al., 2020*). Further, recessive deleterious variation masked by heterosis can similarly cause a signal of increased introgressed ancestry, especially in regions of low recombination (*Kim et al., 2018*). As a result, outlier approaches may have increased rates of false-positive detection of regions under selection. One recent approach modeled local ancestry deviations based on individual-level global ancestry distributions; however, determining a significance threshold for local ancestry deviations remains difficult (*Busby et al., 2017*). Outlier approaches also discard local haplotype information and are not informative about the selection strength or timing. Instead, we developed a suite of methods to identify and characterize selection post-admixture: the *iDAT* summary statistic, application of the SWIF(r) framework to estimate the probability a locus is under selection in an admixed population, and an ABC framework to jointly infer selection strength and initial admixture contributions (Materials and methods).

Methods to detect adaptation driven by alleles introduced through gene flow in human populations have typically focused on ancient admixture between highly diverged populations, often with small contributions from one of the sources (*Racimo et al., 2017*; *Jagoda et al., 2018*; *Setter et al., 2020*). Recent advances leverage patterns of ancestry to consider recent admixture, though perform best for events at least hundreds of generations ago (*Svedberg et al., 2020*). Instead, we emphasize admixed populations as a model for adaptation on historical timescales, with selection dramatically changing genomic variation within tens of generations. This timescale is important for elucidating human history and has implications for conservation genetics and ecology in other organisms. Additionally, our summary statistic approach can be flexibly applied in a variety of inference methods. For example, our implementation in a likelihood-free ABC framework allows for flexible population history models fit to the population of interest. This approach moves beyond identification of loci under selection, allowing joint inference of selection and population history parameters.

We apply these methods to characterize post-admixture adaptation in the Cabo Verdean island of Santiago. The Cabo Verdean populations have a number of advantages for identifying and interpreting selection over the last ~500 years. First, the island geography minimizes within-population structure and provides comparison island populations with shared ancestry components to partially account for demography. Second, historical records give a clear boundary for the earliest onset of selection in the admixed population, based on the initial occupation and admixture in the 1460s (~20 generations in the past). Further, the European and West African source populations have high levels of genetic divergence for human populations, improving local ancestry assignment accuracy. Though errors in phasing or local ancestry assignments are possible and should be considered if applied to other scenarios, it is unlikely that such errors would create signatures as extreme and long ranging as we observe in Santiago.

We inferred that the Duffy-null allele rapidly increased in frequency after admixture as a result of its adaptive resistance to *P. vivax* infection. While we consider this to be the strongest candidate locus, given the large ancestry tracts, it is possible that selection at other nearby loci is responsible for the observed ancestry patterns. The Duffy-null allele shows extreme geographic differentiation, being nearly fixed in sub-Saharan African populations and mostly absent in non-African populations (*Mercereau-Puijalon and Ménard, 2010*; *Howes et al., 2011*; *Gething et al., 2012* ), and multiple populations with ancestry from sub-Saharan Africa show evidence for admixture-enabled adaptation throughout human history at the *DARC* locus (*Hodgson et al., 2014*; *Triska et al., 2015*; *Busby et al., 2017*; *Laso-Jadart et al., 2017*; *Pierron et al., 2018*; *Fernandes et al., 2019*). Further, the Duffy-null allele has well-characterized functional protection against *P. vivax*, a malaria parasite with a documented record of recurrent transmission in Santiago since settlement.

We demonstrated how selection at the *DARC* locus may have affected patterns of ancestry genome-wide (*Figure 5*). The initial admixture contributions inferred under our model were lower than those observed in Santiago today, and we confirmed this pattern more generally in simulations. Population-genetic studies often use loci far from potentially selected sites as putatively neutrally evolving loci, or treat many dispersed loci as neutral based on the assumption that a few

selected sites will not dramatically change genome-wide distributions of summary statistics. However, we found that selection on a single locus may shape patterns of ancestry genome-wide. Individuals with higher genome-wide proportions of ancestry from the source population carrying the beneficial allele are more likely to have a selective advantage in early generations post-admixture as recombination has yet to uncouple global ancestry proportion and the local ancestry of the selected allele (*Pierron et al., 2018*). The genome-wide consequences of adaptation at a single-locus were previously proposed by *Pierron et al., 2018*. However, the scale of simulations was limited, and the extent of this effect on the selected chromosome versus the rest of the genome was not explored. Using whole-genome simulations, we demonstrated that selection on an allele on chromosome 1 increases ancestry proportion on chromosomes 2–22; that is, the increase in global ancestry is not exclusively owing to an increase in ancestry on the selected chromosome (*Figure 5B*). We also showed that this effect can be detected over a range of moderate to strong selection coefficients. Finally, we illustrated how this effect may explain the discrepancy in our observed global ancestry and inferred initial admixture contributions for Santiago (*Figure 5A*). Therefore, since global ancestry is often used to infer initial admixture contributions, these and related demographic inferences may be biased under the common assumption that genome-wide patterns of ancestry reflect demography alone. Importantly, the change in global ancestry will likely have long-term consequences even if adaptation is relaxed. In the absence of negative selection, ancestry is not expected to revert to initial admixture proportions. Instead, drift will not change global ancestry in a specific direction, and the population will evolve neutrally with this new ancestry proportion (*Pierron et al., 2018*).

Difference in observed and simulated global ancestry may be caused by a variety of statistical and evolutionary processes beyond selection, including sampling, estimation method, demographic model misspecification, different rates of migration over time, or drift. Regions aside from *DARC* may also have been under selection post-admixture in Santiago and therefore affected the global ancestry patterns. For example, it is possible there are other genetic associations with *P. vivax* susceptibility; however there are not many published and well-characterized examples (*Zimmerman et al., 2013*). Further, only mutations with large effects, such as the Duffy-null allele, are likely to show significant allele frequency deviations or ancestry-based signatures of selection in just 20 generations.

We also observed a cluster of extreme negative *iDAT* values on chromosome 16 [~chr16: 48,000,000–60,000,000] that may be of interest for future study. The 10 annotated genes in this region and associated gene ontology terms are included in *Supplementary file 1*. A cursory literature search returned no known associations with malarial response in this region, and it is more difficult to draw conclusions on selection history in this region without a prior hypothesis. That said, we note that this region does not show high proportions of West African ancestry compared to the genome-wide distribution (*Figure 1—figure supplement 1*). The extreme *iDAT* values in this region may be influenced by its adjacence to the centromere, which may affect the length of ancestry tracts.

Generally, we suggest that *iDAT* should be used as one line of evidence alongside other summaries of variation, such as those used in our ABC estimation and expected allele frequency calculations. Moreover, in the case of Duffy-null, we had a strong prior expectation for positive selection for the West African haplotype. Analogous to biallelic selection scenarios, with admixture between two source populations, it is difficult to distinguish between positive selection for one ancestry and negative selection for the other ancestry.

Another important limitation for ancestry-based methods of detecting selection in admixed populations is that they are best suited for scenarios wherein frequencies of a selected allele differ greatly between source populations, as is the case with Duffy-null. If an allele is present at similar frequencies in the source populations (i.e. regions of low $F_{ST}$ between source populations), selection will likely affect both ancestries in the admixed population similarly. Future studies into the adaptive histories of admixed populations should consider this limitation on the pool of potentially adaptive variants that can be detected using ancestry-based analyses.

The framework developed here is broadly applicable to detect and characterize selection in other recently admixed human and non-human populations. For most analyses, we assumed a single admixture event 20 generations in the past. The simulation software we use, SLiM, makes it straightforward to consider other models of population history specific to populations of interest for future studies. Further research into the interaction of selection and demography will refine inference.

Additionally, while the timing of admixture in Cabo Verde is well documented, our ABC approach may be extended to infer the timing of selection and admixture as well, as the tract length distribution is informative about these parameters (*Gravel, 2012*; *Liang and Nielsen, 2014*; *Korunes et al., 2020*). Indeed, we demonstrated the need to jointly infer selection and demographic histories.

## Materials and methods

### Genetic data and ancestry inference

For this study, we used SNP array data from *Beleza et al., 2013* which included 564 admixed individuals across the island regions of Cabo Verde. From this data set, we filtered individuals with greater than 5% missing calls overall or greater than 10% missing calls on a single chromosome. This resulted in removal of one individual with high missingness on chromosome 14 (11.36%). We merged genotypes for the remaining 563 individuals with genotypes from 107 IBS (Iberian Population in Spain) and 107 GWD (Gambian in Western Division – Mandinka) samples from high-coverage resequencing data released through the International Genome Sample Resource (*Clarke et al., 2017*; *Fairley et al., 2020*). Our analyses considered autosomal chromosomes only. We selected biallelic SNPs occurring in both the Cabo Verde samples and the reference samples. The final merged data set contained 884,656 autosomal SNPs. Average missingness by SNP was 0.0017.

Using the 884,656 autosomal SNP data set, we performed phasing with SHAPEIT2 using the Phase 3, NCBI build 37 (hg19) reference panel of haplotypes and associated genetic map in IMPUTE2 format (*Delaneau et al., 2013*). Following the SHAPEIT documentation, we first ran SHAPEIT –check to exclude sites not contained within the reference map, followed by SHAPEIT phasing to yield phased genotypes at 881,279 SNPs. For local ancestry inference, we ran RFMix v1.5 on the phased samples using a two-way admixture model (*Maples et al., 2013*). We used the RFMix PopPhased program with default window size, the --use-reference-panels-in-EM option, -e = 2 (2 EM iterations), and --forward-backward. Ancestry references for European and West African source populations, respectively, were IBS and GWD.

We observed a low overall proportion of regions within a given individual's genome assigned as 'unknown' local ancestry by RFMix. The mean and median genome-wide proportion of unknown ancestry in our data set were 0.0089 and 0.0083, respectively. Typically, more than 99% of each individual's genome could be assigned as either West African or European ancestry. Coupled with the fact that there is little historical or genetic evidence of migration from non-West African or non-European populations to Cabo Verde (*Carreira, 1983*; *Verdu et al., 2017*; *Korunes et al., 2020*), we find it is unlikely that individuals in our data set have high proportions of recent ancestry from a non-European or non-African source population.

To validate our phase imputation and RFMix local ancestry calls, we also performed local ancestry assignment using a second method, ELAI, which performs its own independent phasing prior to calling local ancestry (*Guan, 2014*). We ran ELAI under a two-way admixture model, again using IBS and GWD genotypes as references for the source populations. We set the parameters -mg (number of generations) to 20, -s (EM steps) to 30, -C (upper clusters) to 2, and -c (lower clusters) to 10, based on the $5 \times C$ recommendation from the ELAI documentation. The estimates of each individual's average genome-wide ancestry are highly correlated between ELAI and RFMix (Pearson's R = 0.9964; $p < 1 \times 10^{-8}$).

Global ancestry inferred in ADMIXTURE (*Alexander and Lange, 2011*) was averaged over 10 independent runs with randomly chosen seeds using supervised cluster mode with the GWD and IBS individuals specified as the reference populations. Our estimates of global ancestry by individual from ADMIXTURE are consistent with those of RFMix (Pearson's R = 0.9973; $p < 1 \times 10^{-8}$).

The island of Boa Vista was excluded from analyses due to our small sample from the region (26 individuals), leaving a final set of 537 individuals across the three island regions considered in this study (Santiago: 172, Fogo: 129, NW Cluster: 236; *Figure 1A*).

Local ancestry calls can be found at https://doi.org/10.5281/zenodo.4021277.

### *iDAT* score

In order to account for global ancestry patterns that contribute to the ancestry tract-length distribution when identifying loci under selection post-admixture, we developed the *iDAT* score. This

statistic quantifies the length and homozygosity of the tract-length distribution by comparing the decay of tract lengths from alternate ancestries at increasing distance from a site of interest.

First, we describe the Decay in Ancestry Tract (*DAT*) feature, which is calculated similar to expected haplotype homozygosity (*EHH*) (*Sabeti et al., 2002*), using local ancestry tracts rather than haplotypes. For each source population $i$ we calculate:

$$DAT_i(x) = \left(\frac{n_x}{n}\right)^2,$$

where $n_x$ is the number of ancestry tracts that extend some absolute distance $x$ from a position of interest, and $n$ is the total number of ancestry tracts (extending in either direction) that contain the site of interest. We calculate *DAT* at increasing distances from the site of interest. Similar to integrated haplotype homozygosity (*iHH*) (*Voight et al., 2006*), we then calculate the area under the curve for *DAT* as a function of distance from the position of interest, producing *iDAT* (*Figure 2B*). For this study, we calculated *iDAT* only for distances where $DAT \geq 0.25$, that is, where at least half of the ancestry tracts extend that absolute distance from the site of interest. We compare the difference in order of magnitude between *iDAT* for each ancestry, analogous to the integrated haplotype score (*iHS*), we have,

$$iDAT \text{ score} = \ln\left(\frac{iDAT_2}{iDAT_1}\right),$$

where $iDAT_i$ is the *iDAT* calculated for source population $i$. The length of ancestry tracts, and therefore the *iDAT* score, will be influenced by relative ancestry contributions from each source population. So, when possible, we standardized the *iDAT* score using the empirical distribution of *iDAT* scores for 10,000 random positions across the genome. In this way, we deviate from the calculation of standardized *iHS*, which is standardized by the empirical distribution of SNPs with the same allele frequency because the length of haplotypes will be affected by the age of a variant (*Voight et al., 2006*). In the case of admixed populations, we instead need to account for the effect of global ancestry proportion on ancestry tract lengths, where the majority ancestry will tend to have longer contiguous tracts. We standardized by the genome-wide empirical distribution of *iDAT* scores, rather than *iDAT* scores for variants with the same local ancestry proportion, because the variance in local ancestry across the genome can be heavily influenced by drift. That is, to standardize the *iDAT* score, we calculate

$$\frac{\ln\left(\frac{iDAT_2}{iDAT_1}\right) - E\left[\ln\left(\frac{iDAT_2}{iDAT_1}\right)\right]}{SD\left[\ln\left(\frac{iDAT_2}{iDAT_1}\right)\right]}.$$

By standardizing against a genome-wide distribution of *iDAT* values, we can account for demographic parameters, such as admixture proportions and timing since admixture, that may affect the global *iDAT* distribution. Single-locus deviations from the genome-wide expectations may then be indicative of selection at that site, and may warrant further study.

## Single-chromosome simulations

We used SLiM forward simulations with tree-sequence recording to track local ancestry (*Haller et al., 2019*; *Haller and Messer, 2019*). We considered eight different demographic scenarios: combinations of initial population size (N = 10,000 or N = 1000) with either a constant population size or exponential growth at a rate of 0.05 per generation, and either a single pulse of admixture at the start of simulation or continuous admixture at 1% total new migrants per generation (*Table 2*). The proportion of new migrants from each source population was weighted by the respective initial admixture contributions.

For each demographic scenario, we generated 1000 simulations of the human chromosome 1. For realistic recombination rates, we used the population-averaged human genetic map provided by IMPUTE2 (*Delaneau et al., 2013*) (https://mathgen.stats.ox.ac.uk/impute/1000GP_Phase3.html). We simulated admixture from two source populations to form a third admixed population (similar to recipe 17.5 in the SLiM manual) (*Haller and Messer, 2016*). One source population, representing the West African source population, was fixed for a neutral variant at the same position as the Duffy-null

allele (chr1:159174683; GRCh37 coordinates in accordance with genetic map). Because the true West African ancestry contribution is unknown, for neutral simulations this parameter was drawn from a uniform distribution with lower and upper bounds at 0.65 and 0.75, respectively. We simulated the admixed population for 20 generations.

Using the tree-sequence files output from the SLiM simulations, we calculated the five ancestry-based summary statistics for each simulation: West African local ancestry proportion at *DARC*, variance in the distribution of West African local ancestry proportion across SNPs along the chromosome, mean and median West African ancestry tract length containing Duffy-null, and unstandardized *iDAT* score for the Duffy-null variant. We used the unstandardized *iDAT* score because there is no genome-wide distribution of *iDAT* scores for single-chromosome simulations. *iDAT* scores also could not be standardized using the distribution of simulated Duffy-null *iDAT* scores because each simulation had a different starting admixture proportion.

We sampled 172 individuals from each simulation and compared the simulated distribution to the observed values of the statistics for the 172 individuals from Santiago (*Figure 2—figure supplement 2*). The genetic map provided by IMPUTE2 is population-averaged. To determine whether population-specific differences in recombination rate may affect our ancestry-based statistics, we performed the same neutral simulations and comparison to empirical data using one of three genetic maps: GWD or IBS maps from *Spence and Song, 2019*, or an African American (AA) genetic map from *Hinch et al., 2011*. The expectations for mean and median tract length are affected by population-specific differences in recombination rate; however, because recombination rate affects both ancestries equally, the choice of genetic map does not change the expectation for *iDAT* score (*Figure 2—figure supplement 3*). Of note, the AA genetic map contains regions with extremely high recombination rate, resulting in the extreme differentiation in expectation of tract lengths between simulations using this and the other genetic maps, and, in particular, shorter expected tract lengths. Using this map would inflate our estimates of selection strength under the ABC framework; we chose to use the more conservative and general-use IMPUTE2 genetic map for population-averaged recombination rates.

Ancestry tracts extend over large proportions of the chromosome at the timescale of interest for this study (~20 generations). Therefore, in this case, fine-scale recombination rate differences are not expected to significantly affect our expectations for ancestry-based statistics.

## Performance of *iDAT*

The impact of different population size and migration scenarios on *iDAT* is summarized in the Materials and methods, under Single-chromosome simulations (*Figure 2—figure supplement 2*). These scenarios are relevant for the population history of Cabo Verde. Here, in order to understand the general behavior and applicability of *iDAT* for future analyses, we extend the scenarios considered beyond those likely to represent Cabo Verdean history. We consider combinations varying the generations since admixture, the selection coefficient, the initial contribution from the source populations, different chromosome lengths for the position of the adaptive allele, and different *DAT* cutoffs.

First, we considered the demographic history of the admixed population. Modifying the single-chromosome simulations described above, we conducted simulations setting the admixture timing to 10, 100, or 1000 generations in the past and admixture contribution from the source population with the adaptive allele to 0.1, 0.5, or 0.9. This source population was fixed for the variant at the Duffy-null position. We assumed a constant population size of N = 10,000 and a single-pulse of admixture. For each combination of admixture timing and admixture proportion, we generated 1000 simulations for each selection strength of $s = 0$, $s = 0.01$, $s = 0.05$, or $s = 0.1$. This resulted in 36 different scenarios and 36,000 simulations. For each simulation, we calculated *iDAT* for the variant at the Duffy-null position.

To interpret *iDAT* performance under these scenarios, we plot the proportion of Duffy-null *iDAT* values from the selection simulations that are in the bottom fifth percentile of the respective neutral Duffy-null *iDAT* distribution (*Figure 2—figure supplement 4*). We compare within demographic models because admixture proportion has a strong influence on the expectation and possible range of *iDAT* values. We also note that *iDAT* values cannot be calculated when an allele has been fixed in a population, as observed in the older admixture scenarios with high selection strength and high starting admixture proportion from the selected ancestry. As such, this statistic may be more useful

under recent admixture and selection (i.e., fewer than 100 generations in the past) with substantial admixture contributions from both source populations of interest.

We next sought to assess how chromosome size and choice of *DAT* cut-off value affect the expected distribution of *iDAT* values. For this, we performed simulations of human chromosomes 1 (~250 Mb), 7 (~160 Mb), 15 (~100 Mb), and 22 (~50 Mb), to capture a range of reasonable chromosome sizes. We used the associated IMPUTE2 genetic map for each of these simulations. For consistency across simulations, one source population was fixed for a variant at a position at 80% of the physical length of the chromosome. We again assumed a constant population size at N = 10,000 and a single pulse of admixture. Based on the demographic history of Cabo Verde, we assumed the modern-day West African ancestry proportion to be the initial admixture contribution of 0.73 from the source population providing the variant of interest. We simulated the admixed population for 20 generations. We generated 1000 simulations each of neutral ($s = 0$) and strong selection scenarios ($s = 0.05$). For each simulation, we calculated *iDAT* for the tracked variant for distances where $DAT \geq 0.25$ (following our main analyses), $DAT \geq 0.125$, $DAT \geq 0.0625$, and $DAT \geq 0.01$. We show the proportion of *iDAT* values for the simulated variant under selection that are in the bottom fifth percentile of the neutral distribution of *iDAT* values, for each chromosome size and *DAT* cut-off (*Figure 2—figure supplement 5*).

Generally, on this timescale, selection strength, and admixture proportion, *iDAT* performs well across chromosome sizes and cut-off values. That said, we note that the cut-off of 0.25 works slightly better for the smallest human chromosome (chr 22), though we emphasize that its performance is not very different from the other cut-off values. Further, using lower cut-offs requires more computation time, and depending on the SNP density of the data set, this may be an important consideration. We encourage future studies looking to detect signatures of selection using this statistic on smaller chromosomes or different admixture histories, particularly for non-human populations, to consider testing *iDAT* performance under their specific model of interest.

## SWIF(r) implementation

We incorporated ancestry-based summary statistics and admixture simulations into the SWIF(r) framework developed by *Sugden et al., 2018*. For the simulated training set, we followed the single-chromosome simulation framework described above. We considered a single realistic demographic scenario for training: starting population size N = 10,000, exponential growth at a rate of 0.05 per generation, and single-pulse admixture with starting West African ancestry contribution randomly drawn from a uniform distribution from 0.65 to 0.75.

For simulations with selection at a single locus, we assumed an additive selection model ($h = 0.5$). Selection coefficient was randomly drawn from a uniform distribution from 0 to 0.2. We calculated the five ancestry-based summary statistics for each simulation. Under this model, we generated 50,000 neutral simulations and 100 positive selection simulations for training; these training proportions are to reflect a prior probability of selection scenarios at 0.002. Since positive selection is relatively rare compared to neutral scenarios, SWIF(r) calibrates posterior probabilities according to this designated prior probability.

To validate this extension of SWIF(r), we generated a new set of 1000 neutral simulations and 1000 positive selection simulations. *Figure 6* shows a precision-recall plot for a SWIF(r) implementation using the five ancestry-based statistics, a prior positive selection probability of 0.2% (reflecting the training set proportions above), and two classes (neutral or positive selection). *Figure 6—figure supplement 1* demonstrates SWIF(r) performance for each class and across values of admixture contribution and selection coefficient. This SWIF(r) implementation had a low rate of false-negative classification of neutral simulations as positive selection scenarios.

## Inference of selection under a deterministic population-genetic model

To estimate the selection coefficient for the Duffy-null allele based on the dominance and the allele frequency trajectory over 20 generations, we used a deterministic population-genetic model of selection. We used the following recursion equation (*Coop, 2020*):

$$p_{t+1} = p_t + \frac{p_t h s + q_t s (1-h)}{1 - 2 p_t q_t s h - q_t^2 s} p_t q_t,$$

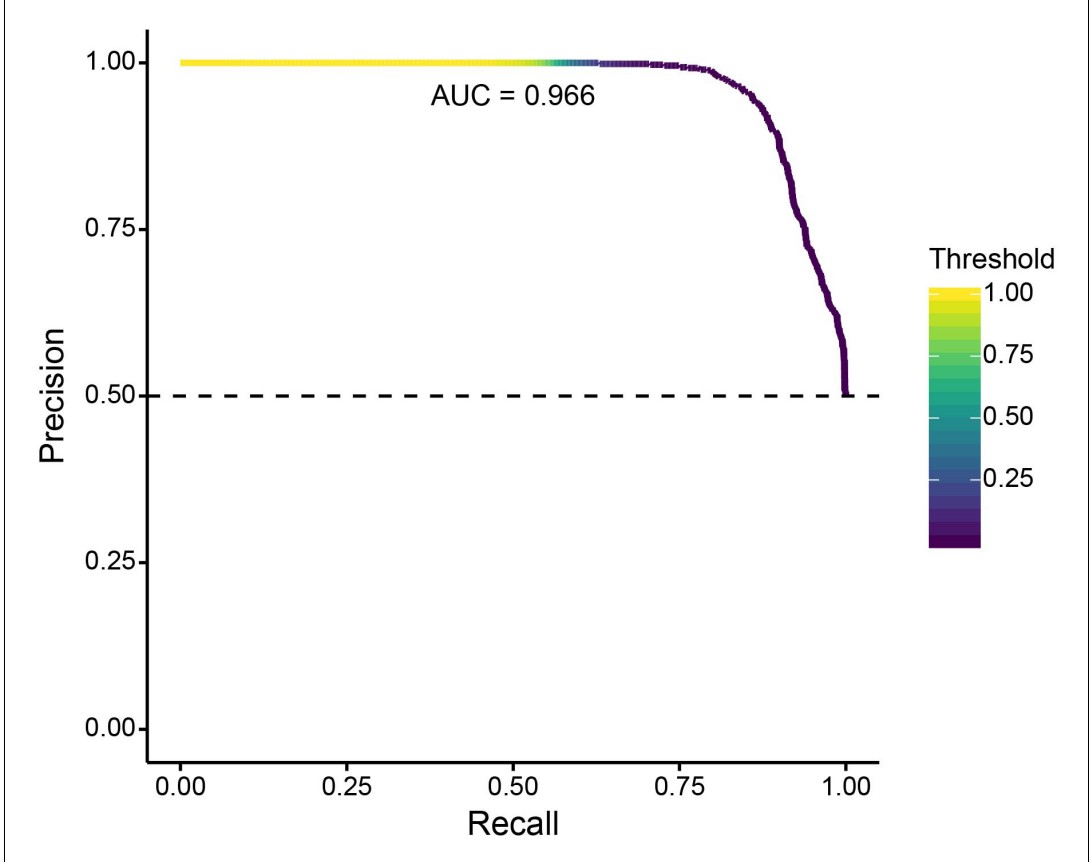

**Figure 6.** Precision-recall curve for validation of SWIF(r) classification of neutral and positively selected variants, using 1000 neutral and 1000 positive selection simulations. With our ancestry-based measures, SWIF(r) achieved an area under the curve (AUC) of 0.966, where an AUC of 1 represents a classifier with perfect skill. Horizontal dashed line indicates the no-skill classifier for this data set.

The online version of this article includes the following figure supplement(s) for figure 6:

**Figure supplement 1.** SWIF(r) classification results for 1000 neutral and 1000 positive selection simulations used for the test set based on Santiago's demographic history.

where $p_t$ is the frequency of Duffy-null in a given generation $t$, $q_t$ is the frequency of the alternate allele in generation $t$, $s$ is the selection coefficient which is constant over time, and $h$ is the dominance coefficient (0 if Duffy-null is dominant, 1 if recessive, and 0.5 in an additive selective model). We calculated the allele frequency over a grid of values for pairs of $h \in [0, 1]$ and $s \in [0, 0.2]$ in 0.005 and 0.001 increments, respectively. **Figure 4A** shows the combinations of $h$ and $s$ that produce $|p_{20} - p_{Duffy}| < 0.01$, where $p_{20}$ is the calculated frequency of the selected allele after 20 generations and $p_{Duffy}$ is the observed frequency of Duffy-null in Santiago (0.834). Because the true initial allele frequency $p_o$ is not known, we performed the analysis for three reasonable starting allele frequencies: $p_o = 0.65$, $p_o = 0.70$, and $p_o = 0.75$.

## Inference of selection coefficient using ABC

We used an ABC framework with nonlinear regression (neural network) adjustment to jointly estimate the posterior distributions for selection strength acting on the Duffy-null allele and the initial West African ancestry contribution (R package '*abc*', with 'neuralnet' method) (*Csilléry et al., 2012*).

We followed the same simulation framework we previously described for our SWIF(r) positive selection simulations of chromosome 1, with the following demographic scenario for training: starting population size N = 10,000, exponential growth at a rate of 0.05 per generation, and single-pulse admixture. We assumed an additive selection model ($h = 0.5$). The selection coefficient ($s$) and the initial West African ancestry contribution ($m$) were drawn from uniform prior distributions,

$s \sim U(0, 0.2)$ and $m \sim U(0.1, 0.9)$. We generated 10,000 simulations for ABC inference and calculated the five ancestry-based summary statistics for each simulation.

We chose the tolerance and hidden layer sizes for ABC estimation based on the best combination of $RMSE$ and $R^2$ values for leave-one-out cross-validation under different combinations of these hyperparameters. We performed cross-validation for 1000 simulations. Estimation accuracy for this data set was similar under a variety of hyperparameter choices. We set the number of units in the hidden layer to 'sizenet=2' and the acceptance rate to 'tol=0.05'. For the cross-validation, our estimates of the selection coefficient had an $RMSE = 0.0083$ and $R^2 = 0.9785$; our estimates of starting admixture proportion had an $RMSE = 0.0090$ and $R^2 = 0.9985$ (*Figure 4—figure supplement 2*).

### Global ancestry simulations

To assess how selection at a single locus impacts genome-wide patterns of ancestry, we simulated changes in global ancestry over 20 generations (*Figure 5*). Whole autosome (22 chromosome) simulations with a realistic recombination map are computationally intensive. Instead, we took two complementary approaches.

First, we performed whole autosome simulations (22 independently segregating chromosomes) in SLiM by specifying a 'crossover rate' between chromosomes of 0.5 per generation. We used the total lengths of autosomal chromosomes from the Human Genome Assembly GRCh37.p13 (https://www.ncbi.nlm.nih.gov/grc/human/data?asm=GRCh37.p13). We used a uniform recombination rate within each chromosome of $1 \times 10^{-8}$ crossovers per base position per generation. We considered the following demographic scenario: starting population size N = 10,000, exponential growth at a rate of 0.05 per generation, and single-pulse admixture. Global ancestry was calculated by taking the mean West African ancestry proportion across the autosome.

Second, we performed two-chromosome simulations (modeled based on human chromosomes 1 and 2) in order to incorporate a human genetic map for realistic recombination rates. We again used the genetic maps provided by IMPUTE2 for chromosomes 1 and 2. Two-chromosome simulations were performed under the same demographic model as whole autosome simulations. Global ancestry was calculated by taking a weighted mean of ancestry for the two chromosomes, where chromosome 2 represented the contribution for the ~92% of the genome that segregates independently from chromosome 1.

To test how selection at a single locus affects global ancestry, we considered both whole autosome simulations and two-chromosome simulations, using a West African ancestry contribution of 0.65 for all simulations in the founding generation. We varied the selection coefficient for the simulated Duffy-null variant on chromosome 1 from 0 to 0.2. We assumed an additive selection model ($h = 0.5$). The results are similar across simulation methods (*Figure 5—figure supplement 1*).

To determine how our ABC estimates of initial West African ancestry contribution differ from final global ancestry after 20 generations of selection at the Duffy-null locus, we next simulated whole autosomes admixing for 20 generations drawing parameters from the previously inferred posterior distributions of the selection coefficient and initial ancestry contribution. Specifically, we used the paired estimates of selection coefficient and West African admixture contribution from each accepted simulation from the ABC analysis and passed those parameters as input for each whole autosome simulation; this produced a distribution of estimates of global ancestry for populations simulated with realistic selection and initial ancestry contributions (*Figure 5A*).

## Acknowledgements

We thank Joshua Schraiber and Kelley Harris for useful discussions. We thank Hua Tang and Greg Barsh for generating genetic data used in this study. We acknowledge support from NIH grant R35 GM133481 to AG and NIH NIGMS grant F32 GM139313 awarded to KLK.

## Additional information

### Funding

| Funder | Grant reference number | Author |
|---|---|---|
| National Institutes of Health | R35 GM133481 | Amy Goldberg |
| National Institutes of Health | F32 GM139313 | Katharine L Korunes |

The funders had no role in study design, data collection and interpretation, or the decision to submit the work for publication.

### Author contributions

Iman Hamid, Software, Formal analysis, Investigation, Visualization, Methodology, Writing - original draft; Katharine L Korunes, Data curation, Software, Formal analysis, Writing - review and editing; Sandra Beleza, Resources, Data curation, Writing - review and editing; Amy Goldberg, Conceptualization, Resources, Supervision, Funding acquisition, Methodology, Writing - original draft

### Author ORCIDs

Iman Hamid (iD) https://orcid.org/0000-0003-2168-9727
Katharine L Korunes (iD) http://orcid.org/0000-0002-2648-4707
Amy Goldberg (iD) https://orcid.org/0000-0001-9306-1539

### Decision letter and Author response

Decision letter https://doi.org/10.7554/eLife.63177.sa1
Author response https://doi.org/10.7554/eLife.63177.sa2

## Additional files

### Supplementary files

• Supplementary file 1. Chromosome 16:46582888–60359576 GO terms. File containing ENSEMBL gene IDs and associated GO terms for the 10 genes that overlap with region showing extreme *iDAT* signatures.

• Transparent reporting form

### Data availability

Scripts for analyses, simulations, and to reproduce figures can be found at https://github.com/agold-berglab/CV_DuffySelection (copy archived at https://archive.softwareheritage.org/swh:1:rev:c8b622c47a4073b7c015998568273c29d2da5836/). Sampling consent forms from original study do not allow for public release of genotype data. Inferred local ancestry information can be found at https://doi.org/10.5281/zenodo.4021277.

The following dataset was generated:

| Author(s) | Year | Dataset title | Dataset URL | Database and Identifier |
|---|---|---|---|---|
| Hamid I, Korunes KL, Beleza S, Goldberg A | 2020 | Rapid adaptation to malaria facilitated by admixture in the human population of Cabo Verde | https://doi.org/10.5281/zenodo.4021277 | Zenodo, 10.5281/zenodo.4021277 |

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
