## [Decision Letter]

**Acceptance summary:**

This paper presents a new method for identifying regions of the genome on which natural selection has acted following recent population admixture. The authors apply it and other approaches to document that the Duffy null allele, which confers resistance to the malaria pathogen *Plasmodium vivax*, has been under strong positive selection in populations of the Cabo Verde of mixed African and European ancestries. The method should be broadly applicable and the specific example is a notable contribution to the understanding of how admixture can enable rapid adaptations

**Decision letter after peer review:**

Thank you for submitting your article "Rapid adaptation to malaria facilitated by admixture in the human population of Cabo Verde" for consideration by *eLife*. Your article has been reviewed by three peer reviewers, one of whom is a member of our Board of Reviewing Editors, and the evaluation has been overseen by Bavesh Kana as the Senior Editor. The following individuals involved in review of your submission have agreed to reveal their identity: Fernando Racimo/Graham Gower (Reviewer #2); George Busby (Reviewer #3).

The three reviewers have discussed the reviews with one another and the Reviewing Editor has drafted this decision to help you prepare a revised submission. They all agreed that a finding of selection on Santiago but not the other two islands provides a nice example of parallel selection on malarial resistance in the recent human past, complementing findings for other African locations. Moreover, they appreciated the development of a new statistical approach to detect selection since population mixture. However, there were a number of concerns raised about the test and its application, as well as some other analyses, and a general feeling that more extensive analyses and explorations of the test performance were needed to build a strong case.

Summary:

This paper reports on selection for the Duffy null allele since humans of African and European ancestry colonized the Cape Verde Islands (~20 generations ago). The Duffy null allele is quasi-fixed in sub-Saharan Africa and confers resistance to *P. vivax*, which was apparently a stronger selection force on one of three islands (Santiago). The authors also document an effect of selection at one locus on genome-wide ancestry (as previously reported by Pierron et al., 2018, cited).

Essential revisions:

1) The new statistical test and its application

i) Most results in the manuscript rely on the phase imputation and local ancestry assignment described here. However, no discussion is given to how errors in the phase-imputation or local-ancestry assignment might influence results (such as calculation of the DAT and iDAT statistics). If any Cabo Verde individuals carry recent ancestry from non-African and non-European sources, what effect might that have on local-ancestry assignment? Was any attempt made to identify such individuals, or verify that none exist in the dataset?

ii) "[W]e calculated iDAT only for distances where DAT>=0.25" One wonders if this choice was made for a specific reason---such as to ensure fairer comparisons across differently-sized chromosomes? For instance, if the DARC locus were instead found on chr22, the ancestry-tract-length distributions might be severely truncated. Some more analyses related to the choice of input parameters to the iDAT scores would help make this statistic more widely applicable and robust to other scenarios, beyond the one studied in this paper. A discussion of when this statistic might be more or less appropriate (could it work with older selection scenarios) would also be a nice addition.

iii) Does the DARC locus show any other selection signals that are not dependent on local ancestry assignments? If not, that would be a good way to further support the use of the iDAT score (in this paper and future studies), in that it could be particularly sensitive to recent selection signals that might not be picked up by other methods. Are there extended haplotype homozygosity signals in the region as well?

iv) Do the authors account for subsequent recent migration from Africa/Europe? Could subsequent pulses of African ancestry explain the observed selection signals? If there has been significant recent migration from Africa into Santiago from Africa, then this will have brought African DARC haplotypes into the population, and driven up the overall West African ancestry proportions. Is this accounted for by the DAT standardisation.

v) There is more African ancestry in general on Santiago c.f. Fogo and the NW Cluster. Indeed, the ancestry proportions on Fogo/NW Cluster are approaching 50/50 African/European. Assuming (as the authors do) that this reflects similar ancestry proportions to the initial source groups, is there power to detect local ancestry change when the ancestries are in roughly similar proportions? Is it possible to delineate what ancestry proportions are best suited to this method? e.g. these power calculations on iHS in Figure 2 from Voight et al., (2006)

vi) How do differences in fine-scale recombination rates between African and European populations affect the test? Perhaps they could define the haplotype length for iDAT in population-specific genetic distance (e.g., using the map from Hinch et al., 2011)?

vii) Are the p-values for iDAT well-calibrated across Fst values? In that regard, and more generally, it would be helpful to see the performance of iDAT for some simulations with real present-day African genomes and European genomes at the estimated starting admixture fractions, in which individuals are mixed and recombination operates in a neutral scenario for 20 generations.

viii) Given the hypothesis that selection should have affected Santiago but not the two other islands, the authors should show that a significant iDAT test is *not* seen for the other two islands.

2) Other targets of selection

i) There are a few other peaks in the selection scan that are not discussed in the text. It would be interesting to see what genes these peaks overlap with and if they have anything to do with the response to malaria; in particular, to explore if the genome-wide ancestry shift is solely due to the DARC locus, or whether other malaria-response alleles could be driving the shift as well.

ii) Are the authors able to estimate whether there has been more selection on African ancestry tracts compared to European tracts? For example, if you sum the iDAT values across the genome, are the results +ve, -ve or 0? If it's -ve, might this be additional evidence that African ancestry has been favoured, over and above being around the DARC locus?

3) Effect of ancestry on other chromosomes

This point is very interesting but hard to understand from what is shown. To see it more clearly, it would be helpful to show the ancestry proportions at Duffy and around it for chromosome 1 versus for the rest of the genome. Moreover, the results of simulations (Figure 4B) are confusing, as it seems counter-intuitive for the effect to be weaker for chr 1 alone than for the whole genome; what might be more readily interpreted is the increase in admixture proportion for chr 1 alone vs for the other 21 autosomes. It would also be important to simulate a single admixture event rather than continuous migration, to evaluate if it would make a difference to the findings.

4) Application of SWIF(r)

It was unclear from the description if the simulations of sweeps on which they trained were of selection in the right demographic setting (i.e., since admixture at appreciable frequency) or using standard sweep from a new mutation in a constant size, random-mating population. If the latter, then the training set for sweeps is not the right one, and the precision recall not informative about the actual problem (which is to distinguish neutral admixture from admixture followed by selection on an allele of one ancestry). On a more minor note, why train on only 100 sweep simulations?

5) The following paper should be cited and discussed: https://www.biorxiv.org/content/10.1101/205252v2

---

## [Author Response]

This paper reports on selection for the Duffy null allele since humans of African and European ancestry colonized the Cape Verde Islands (~20 generations ago). The Duffy null allele is quasi-fixed in sub-Saharan Africa and confers resistance to P. vivax, which was apparently a stronger selection force on one of three islands (Santiago). The authors also document an effect of selection at one locus on genome-wide ancestry (as previously reported by Pierson et al., 2018, cited).Essential revisions:1) The new statistical test and its applicationi) Most results in the manuscript rely on the phase imputation and local ancestry assignment described here. However, no discussion is given to how errors in the phase-imputation or local-ancestry assignment might influence results (such as calculation of the DAT and iDAT statistics).

We agree that errors in phasing imputation or local ancestry assignment may influence results. To test our local ancestry inferences, we called local ancestry using a second independent method: ELAI (Guan, 2014), which performs its own independent phasing in addition to calling local ancestry. Our ancestry estimates are correlated well between methods (Pearson’s R = 0.9964, p<2.2e-16). We include this validation in our Materials and methods. We also mention this possible limitation in the Discussion.

If any Cabo Verde individuals carry recent ancestry from non-African and non-European sources, what effect might that have on local-ancestry assignment? Was any attempt made to identify such individuals, or verify that none exist in the dataset?

Historically, there is no evidence of gene flow from non-African or non-European populations during settlement of the Cabo Verdean islands, and there is little evidence of such in recent years (Carreira, 1983; Korunes et al., 2020; Verdu et al., 2017).

To increase our confidence that no individual in our dataset has significant recent ancestry contributions from a non-European or non-African source population, we calculate the proportion of each individual’s genome that was assigned as “unknown” ancestry by RFMix. This proportion is very low, with the highest proportion of unknown ancestry in an individual’s genome at 0.0157. On average, less than 1% of each individual’s genome was assigned to unknown ancestry. We add these explanations in the Materials and methods.

Additionally, our global ancestry by individual as calculated by ADMIXTURE (which does not depend on phasing or local ancestry assignments) is consistent with the mean of RFMix local ancestry assignments (Materials and methods).

ii) "[W]e calculated iDAT only for distances where DAT>=0.25" One wonders if this choice was made for a specific reason---such as to ensure fairer comparisons across differently-sized chromosomes? For instance, if the DARC locus were instead found on chr22, the ancestry-tract-length distributions might be severely truncated. Some more analyses related to the choice of input parameters to the iDAT scores would help make this statistic more widely applicable and robust to other scenarios, beyond the one studied in this paper. A discussion of when this statistic might be more or less appropriate (could it work with older selection scenarios) would also be a nice addition.

We thank the reviewers for bringing up the important point of validating *iDAT* across various genomic, demographic, and selection scenarios. To this end, we have added a variety of simulations of calculating *iDAT* under different scenarios of admixture timing, initial admixture contributions, chromosome sizes and selection strengths.

Next, we also calculate *iDAT* using various cut-off values, to demonstrate how choices in this parameter value may affect results, or how they might inform future studies that may want to utilize this statistic.

We add a new section to in the Materials and methods and add new Figure 2—figure supplement 4 and Figure 2—figure supplement 5 summarizing results.

In addition to these new analyses, we wish to highlight that we had also performed neutral simulations under various demographic models (e.g. continuous migration, population growth rates, etc.), and our expectations for *iDAT* did not change across these simulations. We move this result to Figure 2—figure supplement 2. We add a mention in the Materials and methods to emphasize this.

iii) Does the DARC locus show any other selection signals that are not dependent on local ancestry assignments? If not, that would be a good way to further support the use of the iDAT score (in this paper and future studies), in that it could be particularly sensitive to recent selection signals that might not be picked up by other methods. Are there extended haplotype homozygosity signals in the region as well?

We agree with the reviewers that demonstrating that a classical selection statistic does not show signatures of selection at *DARC* would strengthen our argument that ancestry patterns are more informative about selection in recently admixed populations. We calculate *iHS* for the ~880k SNPs across the genome in our dataset, for each island. We show that this statistic does not return a significant signal for the Duffy-null SNP (Results and new Figure 3). We also add a discussion that use of this statistic in admixed populations (though common), is likely not informative because the mixing of diverged haplotypes and allele frequencies in the admixture process. Indeed, the higher African ancestry, with higher diversity than European contributions, likely makes *iHS* less likely to detect adaptation.

Additionally, we highlight two results that incorporate information beyond local ancestry as supporting evidence the Duffy region is under selection. First, we turn Table S1 into Table 1, and we make a complementary figure (Figure 1—figure supplement 2) to emphasize a previous analysis comparing allele frequency to global ancestry. We found Duffy is an outlier on Santiago. We move this result up to the Results. Second, we used a deterministic population genetic model of allele frequency trajectories to estimate selection strength for *DARC* (moved to Figure 4A). This model also does not incorporate local ancestry assignments. Such signatures are unlikely, however, to be found in a genome-wide scan without a specific hypothesis that a region is under selection. Therefore, methods such as *iDAT* are important and complementary.

iv) Do the authors account for subsequent recent migration from Africa/Europe? Could subsequent pulses of African ancestry explain the observed selection signals? If there has been significant recent migration from Africa into Santiago from Africa, then this will have brought African DARC haplotypes into the population, and driven up the overall West African ancestry proportions. Is this accounted for by the DAT standardisation.

The reviewers are correct that migration from one of the source populations after admixture would be accounted for by the *iDAT* standardization, which uses the genome-wide distribution of *iDAT* values. We clarify in our Materials and methods that demography will affect *iDAT* values genome-wide, which is why single-locus deviations may be indicative of selection.

We looked at continuous migration (which incorporated recent migration) from African/European source populations for our neutral models (now Figure 2—figure supplement 2 and Table 2). Although continuous migration will affect tract lengths, our 5 ancestry-based statistics all sit outside the expectations for those scenarios.

v) There is more African ancestry in general on Santiago c.f. Fogo and the NW Cluster. Indeed, the ancestry proportions on Fogo/NW Cluster are approaching 50/50 African/European. Assuming (as the authors do) that this reflects similar ancestry proportions to the initial source groups, is there power to detect local ancestry change when the ancestries are in roughly similar proportions? Is it possible to delineate what ancestry proportions are best suited to this method? e.g. these power calculations on iHS in Figure 2 from Voight et al., (2006)

We agree that it is important to demonstrate the performance of *iDAT* under various scenarios. We consider a variety of scenarios, including different ancestry contribution rates, in new simulations with Figure 2—figure supplement 4 and Figure 2—figure supplement 5, with results discussed in Materials and methods. More details in our response to point (ii) above.

vi) How do differences in fine-scale recombination rates between African and European populations affect the test? Perhaps they could define the haplotype length for iDAT in population-specific genetic distance (e.g., using the map from Hinch et al., 2011)?

We thank the reviewers for bringing this up. To address this, we perform neutral simulations of *iDAT* with different population-specific genetic maps. We use IBS- and GWD-specific genetic maps from (Spence and Song, 2019) and the African American-specific genetic map from (Hinch et al., 2011). We compare these results to the population-averaged genetic map provided by IMPUTE2 (Delaneau et al., 2013), which we used for most analyses in our manuscript. We include these analyses in the Materials and methods and added as a new Figure 2—figure supplement 3. We find that although the tract-length expectations under neutrality differ for different population-specific genetic maps, the expectations for *iDAT* do not change, and *DARC* is still an outlier. This is because the recombination rates will equally affect the European and West African ancestry tract length patterns.

Further, on this time-scale, ancestry tracts extend over many megabases. Therefore, we don’t expect differences in fine scale recombination rates to significantly affect our expectations for our ancestry-based statistics. We add this discussion to the Materials and methods.

Interestingly, we find that there are many regions in the African American genetic map from Hinch et al., that have extremely high recombination rates, and we are doubtful that these rates of crossover are biologically feasible. However, the high rates make *DARC* even more of an outlier, so the combined IMPUTE2 map we used is more conservative.

vii) Are the p-values for iDAT well-calibrated across Fst values? In that regard, and more generally, it would be helpful to see the performance of iDAT for some simulations with real present-day African genomes and European genomes at the estimated starting admixture fractions, in which individuals are mixed and recombination operates in a neutral scenario for 20 generations.

The reviewers ask an important question about whether *iDAT* can perform well when allele frequencies in source populations are similar. Our expectation is that no ancestry-based statistic will be highly informative in regions with low *Fst* between the source populations, because the two ancestries will be equally affected by selection. This is an important, and perhaps often overlooked, point when conducting standard ancestry-outlier scans as well. We include discussion of this limitation in the Discussion. Specifically, we emphasize that *iDAT* (and other ancestry-based signatures) are useful when allele frequencies are substantially different between source populations, which is the case with Duffy-null.

viii) Given the hypothesis that selection should have affected Santiago but not the two other islands, the authors should show that a significant iDAT test is *not* seen for the other two islands.

We thank the reviewers for pointing this out. We move the genome-wide *iDAT* results for all islands to be Figure 2—figure supplement 1, to emphasize this analysis.

2) Other targets of selectioni) There are a few other peaks in the selection scan that are not discussed in the text. It would be interesting to see what genes these peaks overlap with and if they have anything to do with the response to malaria; in particular, to explore if the genome-wide ancestry shift is solely due to the DARC locus, or whether other malaria-response alleles could be driving the shift as well.

The reviewers refer to a peak in *iDAT* on Chromosome 16 around the physical positions ~48000000-60000000. Using Ensembl’s BiomaRt feature, we find that there are 10 annotated genes that overlap with that region. Using the PANTHER Gene Ontology search software (http://geneontology.org/), we generate a list of GO terms for those genes (Added as Supplementary file 1). We perform a brief literature search for the genes in that peak *iDAT* region, and find no hits for published association with malarial response. We include these notes in the Discussion.

Further, West African ancestry proportion in this region is not an outlier (newly added Figure 1—figure supplement 1). This region sits adjacent to the centromere on chromosome 16, so it is possible that this *iDAT* result is driven by genomic features rather than selection. In general, we suggest using *iDAT* together with other summaries of variation, such as in our ABC simulations and allele frequency calculations. We clarify this in the Discussion.

ii) Are the authors able to estimate whether there has been more selection on African ancestry tracts compared to European tracts? For example, if you sum the iDAT values across the genome, are the results +ve, -ve or 0? If it's -ve, might this be additional evidence that African ancestry has been favoured, over and above being around the DARC locus?

Yes, the sum of unstandardized *iDAT* values for Santiago is negative, but this is true of the other islands as well (though not as extremely negative). This is likely more indicative of higher admixture contribution from the West African source population than selection for African ancestry overall. Indeed, we compared *iDAT* to the genome-wide background because the range of values is dependent on admixture contributions.

In general, for two source populations, it is difficult to differentiate positive selection for one ancestry from negative selection for the other; however, we have a strong prior expectation of positive selection for African ancestry in this case. We now note this in the Discussion.

3) Effect of ancestry on other chromosomesThis point is very interesting but hard to understand from what is shown. To see it more clearly, it would be helpful to show the ancestry proportions at Duffy and around it for chromosome 1 versus for the rest of the genome. Moreover, the results of simulations (Figure 4B) are confusing, as it seems counter-intuitive for the effect to be weaker for chr 1 alone than for the whole genome; what might be more readily interpreted is the increase in admixture proportion for chr 1 alone vs for the other 21 autosomes. It would also be important to simulate a single admixture event rather than continuous migration, to evaluate if it would make a difference to the findings.

We agree with the reviewers that our old Figure 4 (now Figure 5) was difficult to follow. As such, we change Figure 5B to show the mean ancestry on chromosome 1 (black circles) and the mean ancestry on the other 21 autosomes (grey circles); both show a positive relationship with increasing selection coefficient. We changed the explanation in the Results and Figure 5B legend to reflect this.

We clarify in the text that we simulate under a single admixture model for this analysis (Results; Materials and methods). Our aim is to demonstrate a proof-of-concept, rather than inference of ancestry under a model.

4) Application of SWIF(r)It was unclear from the description if the simulations of sweeps on which they trained were of selection in the right demographic setting (i.e., since admixture at appreciable frequency) or using standard sweep from a new mutation in a constant size, random-mating population. If the latter, then the training set for sweeps is not the right one, and the precision recall not informative about the actual problem (which is to distinguish neutral admixture from admixture followed by selection on an allele of one ancestry). On a more minor note, why train on only 100 sweep simulations?

For our SWIF(r) simulations, we used the same demographic model as used for the majority of our other analyses (i.e. an island model, with the beneficial allele fixed in one source population prior to admixture, and selection starting with admixture at appreciable frequencies). We believe this may have been confused by our use of the word “sweep”. We change this to “positive selection” or “selection scenario” throughout the main text and in supplemental figure labels associated with SWIF(r) validation.

We also add text to clarify that we used 100 “selection scenario” and 50k “neutral scenario” simulations for training to reflect our prior probability proportion of selection:neutral 0.002:1 (Materials and methods). We expect selection of our type to be relatively rare across the genome, and SWIF(r) calibrates its posterior probabilities according to the prior probability. So, our training set reflected these proportions.

5) The following paper should be cited and discussed: https://www.biorxiv.org/content/10.1101/205252v2

We thank the reviewers for pointing out this study, it is highly relevant. We added a citation of this article in the Results wherein we discuss past studies that have used local ancestry deviations to identify potential regions under selection. We also include it in the Discussion wherein we cite past studies which have identified *DARC* as a region potentially under selection in recently admixed populations. Finally, we specifically mention this article in the Discussion.